

# Exploring the partial use of the Mo.S.E. system as effective adaptation to rising flood frequency of Venice

Riccardo A. Mel[1]

[1]Department of Environmental Engineering, University of Calabria, Rende (CS), 87036, Italy

5  *Correspondence to*: Riccardo A. Mel (riccardo_alvise.mel@unical.it)

**Abstract.**

In times of climate change the impact of coastal hazards should be mitigated by identifying and implementing effective adaptation strategies, encompassing a balanced mix of structural and non-structural measures based on high level scientific knowledge. Due to its hydro-geological features, the Venice lagoon (Italy) is particularly vulnerable to climate change. Some 10 structural measures have been adopted over time to protect Venice from flooding, among which a system of flap gates (Mo.S.E. system) has been operational under testing phase since October 2020. However, relative sea level rise and wind setup pose relevant management challenges, as a frequent closing of the lagoon would have negative impacts on flushing capacity, fishing industry and port activities. Hence, optimal operation rules for the existing control structure are searched to anticipate and to adapt to a possible acceleration of sea level rise induced by climate change. Here, the focus is on the hydrodynamic effects of 15 a partial closure of the Mo.S.E. barriers that, with respect to closing all the three inlets of the Lagoon, could play a role in reducing the economic and environmental impacts of the Mo.S.E. system. The main goal is to identify the flooding events that can be counteracted by closing only the Lido inlet, which is the closest to the city of Venice. Based on the tidal and meteorological dataset collected in the period 2000-2019, a robust modelling exercise indicates that the closing of the Lido inlet only would protect the Venice lagoon from two third of the flooding events up to a relative sea level rise of + 0.4 m.

## 1 Introduction

Floods are a significant long-term risk to society (Wadey et al., 2012). While the population generally adapts to them (Van Koningsveld et al., 2008; Dawson et al., 2011; Jongman, 2018), this is mostly accomplished as consequence of real events (Mel et al., 2021b, under review), which can involve significant economic losses and deaths (Hinkel et al., 2014; Jongman et al., 2014; Paprotny et al., 2018). In Europe coastal flooding is one of the most threatening natural hazard (e.g. Capobianco et 25 al., 1999; Battjes and Gerritsen, 2002; Liquete et al., 2013; Drejza et al., 2019), and flood risk is expected to further increase in the near future in all the acting components, as sea level (SL) rise, tides, waves, and storm surges (Semedo et al., 2013; Wang et al., 2014; Alfieri et al., 2015; Androulidakis et al., 2015; Marcos et al., 2012; Vousdoukas et al., 2017; Jongman,



2018), with particular reference to extreme events (e.g. Haigh et al., 2011; Menéndez and Woodworth, 2010; Wahl et al., 2011; Woth et al., 2006; Pranzini et al., 2015; MedECC, 2020). Coastal flood hazard is exacerbated by climate change, anthropogenic

modifications of the land and of the environment, and by socio-economic factors (Kundzewicz et al., 2014; Pycroft et al., 2016), driving morphological changes and erosion processes (Ciavola et al., 2011), as well as coastal protection failures (Oumeraci, 1994; Matias et al., 2008; Vousdoukas et al., 2017), reducing the effectiveness of the traditional defence strategies (Brown et al., 2005; Few et al., 2007; OECD, 2016). As the flooding risk can never be entirely prevented fixed (Diez et al., 2011), effective adaptation strategies are needed (McCarthy et al., 2001; Klein et al., 2001; Hinkel et al., 2014; Mechler et al.,

2014; Cutter and Gall, 2015; Diaz, 2016), based on a diversified approach of interventions, which may include structural and non-structural measures aimed at preserving the social, economic and environmental functions of coastal areas (Aerts et al., 2014; Lincke and Hinkel, 2018; Jongman, 2018; Tiggeloven et al., 2020). Structural measures (i.e., hard and soft flood defence works designed for long-term operating, see Maiolo et al., 2020) should be supported by non-structural measures, comprising laws, policies, regulations, planning instruments, optimal management protocols, risk assessment and informational systems

for coastal ecosystems and resources management (Smith and Lenhart, 1996; Smit and Wandel, 1999; Kundzewicz, 2002; Moser et al., 2018; De Alencar et al., 2020; Elko and Briggs, 2020; Mel et al., 2020). Although several approaches and models have been proposed to optimize planning and management actions, implementing effective strategies requires complex decision-making processes and an innovative approach to flood defence (Few et al., 2007; Tobey et al., 2010), addressing the major institutional, economic and environmental barriers (EC, 2009; Ward et al., 2017; Molinaroli et al., 2018).

Floods are a recurrent hazard for the iconic city of Venice (Mel and Lionello, 2014), a clear example of a huge historical, cultural and environmental heritage at risk. Flood frequency has significantly worsened since the 1950s (Battistin and Canestrelli, 2006; Lionello et al., 2012; Mel et al., 2019a; Lionello et al., 2021) due to relative sea level rise (RSLR) which is caused by local subsidence and decadal-scale acceleration in the rate of global sea level rise driven by climate changes (Carbognin et al., 2004; Ferla et al., 2007; Lionello, 2012; IPCC, 2013; Zanchettin et al., 2020). This alarming trend is expected

to continue and even accelerate within the present century (IPCC, 2013). The detrimental effect produced by storm surges in the Adriatic Sea has been shown by the events of 4 November 1966 (De Zolt et al., 2006; Lionello et al., 2021) and 12 November 2019 (Cavaleri et al., 2020; Ferrarin et al., 2020) which produced enormous loss and damage to the historical buildings, economic activities in the coastal settlements located within the Venice lagoon, and even deaths.

The resilience of the Venice lagoon has been permanently linked to the human presence since remote times. The Venetians

altered the environment by building several hard structures, with the aim to preserve economic interests, for defence purposes and to maintain the very existence of the lagoon (Molinaroli et al., 2009; D'Alpaos, 2003; Sarretta et al., 2010; Ferrarin et al., 2013). As non-structural measures adopted for prevention, preparedness and response to floods, the Venetians raised the ground floors, placed steel barriers at the entrance to buildings and adapted the existent electrical systems (Indirli, 2014; Molinaroli et al., 2018). Extreme storms were catalysts of the awareness rising that the Lagoon had lost much of its natural



resilience to floods (Samiolo, 2012), highlighting the need to build new flood defence structures and to invest more in adaptation strategies (Munaretto et al., 2012). After the 1966 flood, the Italian government supported by the scientific community and supranational organisations like the UNESCO (UNESCO, 1969) promoted several structural measures to reduce the impact of flooding, erosion process and water pollution (e.g. Italian Law n. 171 of April 16, 1973; Italian Law n. 798 of November 29, 1984), as the Experimental Electromechanical Module (Mo.S.E.), a system of mobile barriers aimed at

closing the inlets temporarily to protect the Venice lagoon through a set of 78 independently flap gates (Eprim, 2005; Trincardi et al., 2016).

In response to the devasting flood event occurred in 2019, safeguarding of the Venice lagoon became a pressing priority. Works to build the Mo.S.E. system, begun in 2003. Since 3 October 2020, the Mo.S.E. system has been operating under testing phase (see Mel et al., 2021b, under review). Within the first three months, the Mo.S.E. system was operated 20 times during

high tide conditions (see Appendix A for the detail of the operations). However, poor meteorological forecast, novel management options, economic and environmental impact of the closures rose a vast debate about the optimal use of the system. According to recent studies, the use of the Mo.S.E. system on the one hand contributes to lowering the short term monetary costs induced by the periodical flooding (Nunes et al., 2005); on the other hand, if implemented too frequently or for too long a time, it could affect the water quality of the lagoon (Melaku et al., 2001; Viero and Defina, 2016), and induce

additional direct and indirect costs to fishing and port activities (Costa, 1993; Chiabai and Nunes, 2008; Vergano et al., 2010). As the need to close the barriers becomes more frequent in view of the ongoing processes of land subsidence and RSLR, the optimal management of the Mo.S.E. system will play a crucial role in balancing the needs of flood protection with the issues related to port activities and ecosystem preservation. According to Umgiesser (2020) and Mel et al. (2021a), only + 0.3 m of RSLR would generate almost 100 closures/year, often involving multiple tidal cycles, and disrupting a significant portion of

vessel traffic (Brotto and Gentilomo, 1998; Umgiesser and Matticchio, 2006). In addition, the flushing of the lagoon would be greatly reduced, building up toxic substances and damaging its delicate ecosystem (AGU, 2002; UNESCO, 2011). In this context, a possible adaptation strategy consists in limiting the extent of the closure of the inlets through a partial use of the Mo.S.E. system, (i.e., the closure of one or two inlets at a time or a partial closure of each inlet). In particular, the closure of the Lido inlet only can mitigate the flood hazard without significantly affecting the water quality and the economy of the

Lagoon, since it is scarcely used for commercial traffic, as the Malamocco inlet, or fisheries, as the Chioggia inlet. Hereinafter, such partial operation of the Mo.S.E. system is named as Partial Closure of the Lagoon involving the Lido inlet only (PCL). Some recent studies that attempted to quantify the effects of the partial operation of the Mo.S.E. system (Cavallaro et al., 2017; Umgiesser et al., 2020) are affected by important limitations: the sea level thresholds associated to the partial closure of the barriers were kept fixed and equal to the thresholds associated to the full Mo.S.E. closure (Eprim, 2005), and the role of tidal

range, wind setup and intragate infiltration was neglected. The present contribution aims to fill this knowledge gap by a systematic numerical investigation of the effects of the PCL on the hydrodynamic of the lagoon. Specifically, the main goals



are (a) identifying a relationship between tidal range and reduction of the SL peak at the main settlements of the Venice lagoon, namely Venice, Burano, and Chioggia; (b) investigating the effects of wind setup and intragate infiltration on such relationship; (c) address the potential benefit of the PCL in reducing the frequency of full Mo.S.E. closures in the present and future

scenarios. The results of the study can be beneficial for research and decision-making efforts toward an optimal management of the Mo.S.E. system.

## 2 Materials and method

### 2.1 Venice lagoon and Mo.S.E. system

The Venice lagoon, stretching for more than 60 kilometres from north to south and about 12-16 kilometres from west to east,

is the largest Mediterranean brackish water body. It is connected to the Adriatic Sea through three inlets, namely, from north to south, Lido, Malamocco and Chioggia, whose widths range from 400 m to 800 m and depths between 6 m and 14 m. Note that the Lido inlet is divided in two parts by the Mo.S.E. infrastructure, named "Treporti canal" and "San Nicolò canal", see Fig. 1. The Lagoon, characterized by an average water depth of about 3 m, consists of a complex system of channels, tidal flats, salt marshes and small islands, and a high heterogeneity in physical, biogeochemical, and biological conditions of

mutually interacting habitats (Carniello et al., 2009; Finotello et al., 2018; Molinaroli et al., 2018). The most prominent urban settlements are Burano, in the northern lagoon, Venice in the central lagoon, and Chioggia, located in the southern lagoon (Fig. 1). The main circulation forcing factors are a semidiurnal tidal regime and the wind. The average residence time ranges from 1 day for the areas located close to the inlets to 30 days for the inner areas (Viero and Defina, 2016). Nutrient and pollutants mostly originate from urban areas and from the drainage basin (Melaku et al., 2001). For each tidal cycle, the water exchanged

through the inlets is about a third of the total volume of the lagoon (Gacic and Solidoro, 2004). In calm atmospheric conditions, when circulation is influenced only by the tide, the Venice lagoon is subdivided into three sub-basins, one for each inlet, separated by two watersheds through which the residual flow is minimum (D'Alpaos, 2003; Solidoro et al., 2004). The propagation of the tide is affected by the shape of the inlets and the morphology of the lagoon. The tidal range reduces proceeding from the inlets towards the lagoon interior, showing a progressive propagation lag. These alterations in tidal signal

produce significant lagoonal currents, triggered by the prevailing SL differences between different regions of the same sub-basin. Prevalent winds are north-easterly (Bora wind) and south-easterly (Sirocco), of speed up to 20-25 m/s (Mel et al., 2021a).

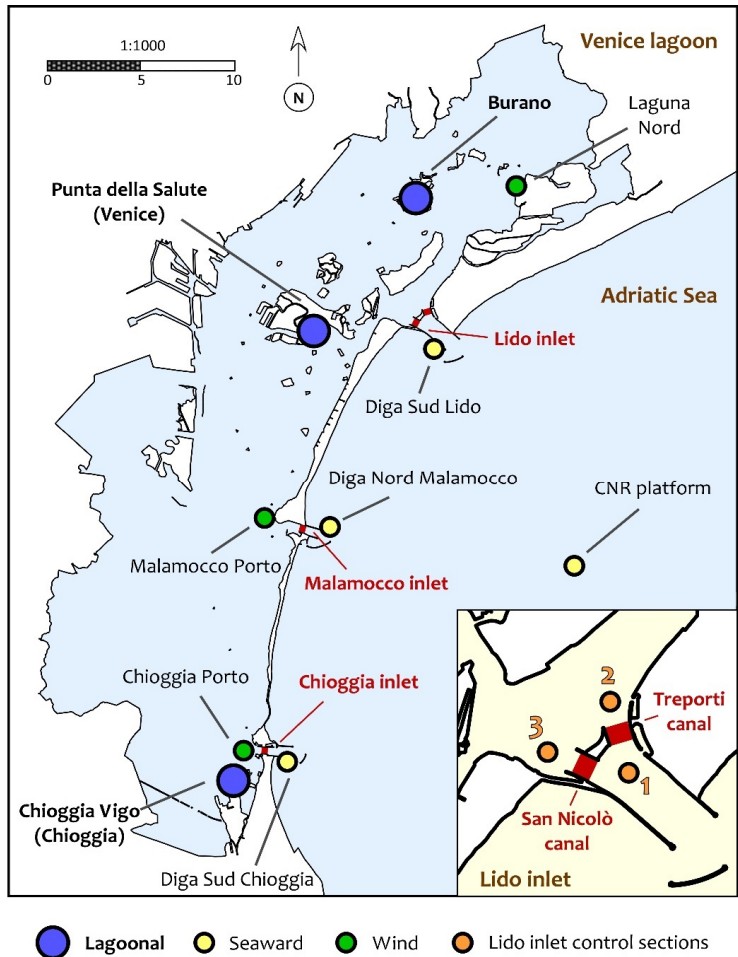

**Figure 1:** The Venice lagoon with the location of the Mo.S.E. barriers (thick red lines). The bullets show the location of wind (green) and tide stations (blue, lagoonal gauges; yellow seaward gauges) used in the present work, coming from the database of the Tidal Forecast Centre
of the Venice Municipality (Centro Previsioni e Segnalazioni Maree, CPSM).

For more than a millennium, Venice has endured by altering the environment of the lagoon (Brambati et al., 2003). Several hydraulic works, such as river diversion, building of large sea wall, digging new navigation canals, enabled the historical heritage of Venice and the related industrial and economic activities to survive, producing a significant impact on the morphology of the lagoon and on the local tidal regimes (Carniello et al., 2009; D'Alpaos, 2010; Sarretta et al., 2010; Silvestri



et al., 2018). By the 1970s more than half of the lagoonal surface had been reclaimed or cordoned off for business-related purposes, altering the natural morphological features and amplifying the tidal range (Molinaroli et al., 2018). As from the very beginning Venice has co-existed with the sea by adopting several measures to adapt to the flood hazard, at present the Mo.S.E. system is conceived to protect the lagoon by temporarily separating it from the sea during the flood events. The Mo.S.E. system consists of four separate storm surge barriers (Fig. 1) formed by some independent flap gates 20 m wide, 5 m thick and between

18 and 28 m high, hinged on the bottom of the inlets. The Lido inlet has two barriers with 21 (Treporti canal) and 20 (San Nicolò canal) elements, Malamocco and Chioggia a single barrier of 19 and 18 elements respectively (see https://www.mosevenezia.eu/progetto/ for more technical details). In normal tidal conditions, the flap gates rest full of water on the bottom of the inlets. When a tide is expected to exceed the safeguard threshold, compressed air is pumped into the gates, allowing the barriers to rotate upwards interrupting the tidal flow (Gentilomo and Cecconi, 1997). The safeguard threshold,

defined as the SL that should not be exceeded during high tides (Umgiesser, 2020), is set to 1.10 m above the local datum at Venice and Burano and 1.30 m at Chioggia, which is protected since 2012 by a local defence installed at both ends of the main canal crossing the historical centre. As an important note, during the present testing phase of the Mo.S.E. system no specific management protocols have been adopted. The barriers have been operated partially and/or asynchronously in order to estimate the time needed to raise and drop the gates, monitor the hydrodynamics of a regulated lagoon and highlight possible

shortcomings (see Appendix A for the details of each operation).

In this work, SL elevations refer to the official local datum of the Punta della Salute (PS) SL gauging station, located in Venice city centre, whose zero is located 0.23 m below the national vertical level datum (IGM 1942) and 0.34 m below the present mean SL (Cavaleri et al., 2020).

### 2.1.1 Data and tidal characteristics

The SL, wind speed and wind direction data used in the present work come from the database of the Tidal Forecast Centre of the Venice Municipality (i.e., Centro Previsioni e Segnalazioni Maree, CPSM). A first dataset consists in two years (2019 - 2020) long records of SL, wind speed and direction collected every ten minutes, used to reproduce some recent flood events (all the gauges are reported in Fig. 1). From the same institution, SL peaks and troughs gauged within the period 2000-2019 (including a Metonic cycle) define the scope of the present work in terms of tidal range during the flood tide (i.e., the SL

difference between the peak and the previous trough, hereinafter named as tidal range) and of tidal semi-period during the flood tide (hereinafter named as tidal period). Mean tidal range at PS is 0.60 m, with more than 20% of the data greater than 0.8 m (i.e., spring tide, see Fig.2a). Tidal period shows a lower variability of the distribution, with a mean value at PS of about 6h 20' and almost 90% of the data located between four and eight hours (Fig. 2b).

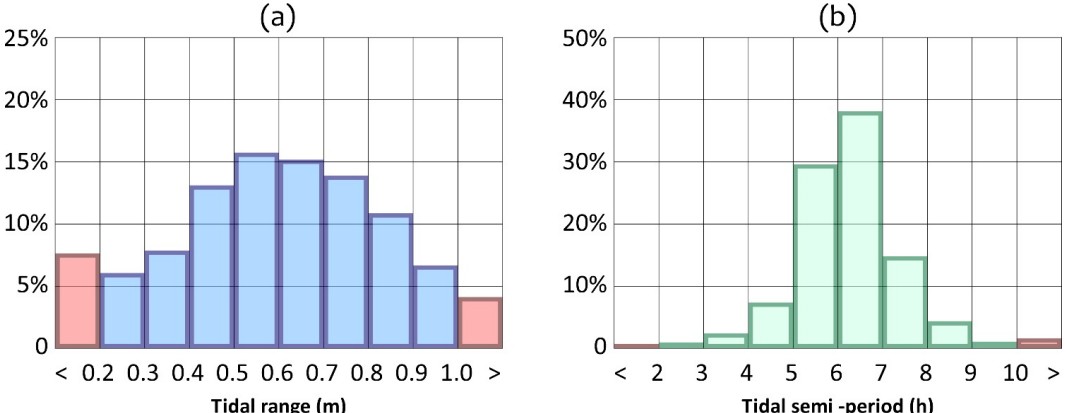

**Figure 2:** Dataset 2000-2019, Punta della Salute (PS). Frequency distribution of tidal range (a) and tidal semi-period (b). Data refer to flood tide.

### 2.2 Hydrodynamic model

Numerical simulations were carried out using the WWTM (Defina, 2000; Carniello et al., 2011), a two-dimensional mathematical model developed at the Department of Civil, Environmental and Architectural Engineering, University of Padova and successfully applied in the Venice lagoon and in other tidal environments (Carniello et al., 2005; Mariotti et al., 2010; Zarzuelo et al., 2018; Mel et al., 2019b; Pivato et al., 2020). WWTM is a coupled wind wave-tide model that solves the full shallow water equations using a semi-implicit staggered numerical scheme based on the Galerkin's approach (Martini et al., 2004; D'Alpaos and Defina, 2007). The finite element method allows to employ elements with different shapes and sizes, providing high flexibility in the spatial discretization of complex geometries, typical of shallow water basins. WWTM describes the hydrodynamic flow field including the flooding and drying processes using a statistical, physics-based approach (Defina, 2000). The wind-wave generation and propagation are computed by solving the equation of wave action conservation, parameterized using the zero-order moment of the wave action spectrum in the frequency domain. The spatial variability of the wind field is accounted by adopting the interpolation technique proposed by Brocchini et al. (1995). WWTM has been used to reproduce tide propagation, wind setup and the effect of the closure of the Lido inlet (i.e., the Treporti and San Nicolò canals, see Fig. 1) by reproducing the raising of the flap gates of the Mo.S.E. system. The computational mesh used in this study reproduces the Venice lagoon and a portion of the Adriatic Sea in front of the three inlets, for a total area of about 600 km². It consists of about 51,000 nodes and 97,000 triangular elements of side-length of about 100 m for tidal flats and 10 m at the three inlets, where the spatial gradients of the velocity are significant even for small tidal ranges. The elevation of the computational elements is assigned on the basis on the most recent (2012) bathymetry of the Venice lagoon provided by the





Venice Water Authority. The Strickler bed roughness coefficient (Ks) has been calibrated (Carniello et al., 2005; 2011) in order to represent the energy dissipation over the different, morphologically representative, portions of the lagoon. It is worthwhile pointing out that the capability of the model to reproduce the tide propagation within the lagoon and the flow rates across the three inlets has been widely tested by comparing WWTM results to data records (e.g. Carniello et al., 2011; Mel et al., 2019b), showing absolute errors generally comparable to measurement precision (see Appendix A for the simulation of all

the events that involved the raising of the Mo.S.E. barriers occurred from 3 October 2020 to 11 February 2021).

In the present work WWTM is forced by imposing the tidal signal at the three inlets (i.e., Diga Sud Lido, Diga Nord Malamocco and Diga Sud Chioggia). The spatial and temporal distribution of the wind field is reconstructed following the approach described in Carniello et al. (2012) and considering the wind climate gauged at the Laguna Nord, Malamocco Porto and Chioggia Porto (see Fig. 1). The wind shear stress at the water surface has been widely calibrated and tested against data

collected during the most prominent storms, including three recent events occurred when the Mo.S.E. system was operational (Mel et al., 2021b, under review).

### 2.3 Simulation set-up

The effect of the PCL has been addressed on three different scenarios, namely scenario (I), (II) and (III). In the scenario (I), the simulations were designed to reproduce tidal circulation with no wind, and the only forcing was a synthetic sinusoidal tide

imposed at the three inlets. Different tidal ranges (from 0.1 m to 1.4 m) and different periods (from 4 h to 8 h) were investigated. All the forcing tides are characterized by a SL peak of 1.20 m, given that the PCL would be effective for peaks from 1.10 m (the safeguard threshold of PS and Burano) to 1.30 m (the safeguard threshold of Chioggia, which would marginally benefit from the closure of the Lido inlet). In the scenario (II), constant velocities (from 5 m/s to 20 m/s) of north-easterly wind (i.e., Bora, incoming direction of 45°N) and south-easterly wind (i.e., Sirocco, incoming direction of 165°N) have been

superimposed to the same tidal forcings of scenario (I). The scenario (III) comprises all the events (n° 42) occurred in the period 2019 – 2020 with gauged SL peak ≥ 1.10 m at PS, including some events when the Mo.S.E. system was operational (in such cases the safeguard threshold has been referred to the SL gauged at CNR platform). In this scenario, the model has been forced with the SLs gauged at the three inlets and the wind measured at Laguna Nord, Malamocco Porto and Chioggia Porto (Fig. 1). These events are characterized by different tidal ranges (from 0.3 to 1.5 m), different periods (from 3 to 8.5 hours,

plus two events that involved two tidal cycles), different meteorological conditions (i.e., wind speed and direction), and different initial conditions (i.e. SL at the troughs and tidal range of the previous tidal cycle).

For all the three scenarios, and for each event, the tidal dynamics has been reproduced through four different runs, namely:

- run (a), without any operation at the Mo.S.E. barriers (unregulated lagoon);
- run (b), by closing the Treporti and San Nicolò canals at the beginning of the flood tide, i.e. when the flow rate
through the Lido inlet computed in run (a) is null;


- run (c), as run (b), but including the intragate infiltration through the Treporti and San Nicolò barriers estimated by using a classical formulation for orifice flows as proposed by Mel et al. (2021b, under review), on the basis of the SL difference computed in run (b) between control section 1 and 2 (Treporti) and between control section 1 and 3 (San Nicolò) (Fig. 1);

- run (d), as run (c), but opening the Treporti and San Nicolò canals when the SL difference computed in run (c) between control section 1 and 2 (Treporti) and between control section 1 and 3 (San Nicolò) is null.

The effect of the PCL has been addressed by comparing the SLs peaks computed at PS, Burano, and Chioggia between run (d) and (a).

## 3 Results and discussion

### 215  3.1 Scenario (I): reduction of the sea level peak

Figure 3 shows the reduction of the SL peak (i.e., the SL peak difference between run (a) and (d), hereinafter named as $\Delta H$) due to the PCL under the scenario (I), i.e., a sinusoidal tide with no wind. Panels (a), (b) and (c) show the $\Delta H$ at PS, Burano, and Chioggia, for a forcing tide characterized by a period of 6 hours and different tidal ranges. A first, important, result, is the linearity of the relationship between tidal range and $\Delta H$ at the three gauges. Notably, a linear relationship is achieved only if

the tidal range is referred to the tide signal imposed at the inlets (solid bullets, hereinafter named as seaward tidal range, $H_{SW}$) and not at the specific gauge (opaque bullets, hereinafter named as local tidal range). This can be explained by effect of tide propagation on the local tidal range, which can be significantly altered far from the inlets. The benefit of the PCL in terms on $\Delta H$ is significant:

$$\Delta H_{PS} = 0.20 \cdot H_{SW} \quad (1)$$

$$\Delta H_{BURANO} = 0.27 \cdot H_{SW} \quad (2)$$

$$\Delta H_{CHIOGGIA} = 0.04 \cdot H_{SW} \quad (3)$$

For a tidal range of 0.6 m, the PCL would reduce the SL peak of about 0.12 m at PS, 0.16 m at Burano and 0.02 m at Chioggia. Results are not affected by the tidal period, as shown in Fig. 3d for the period range of 4 – 8 hours. Drawing a linear relationship is fundamental for investigating the effectiveness of the PCL on long dataset, as the $\Delta H$ can be estimated through the SL

difference gauged between the SL peak and the previous trough. As a note, the contribute of the flow rate discharged into the lagoon due to leakage (run (c)) through the gates on the relationships (1), (2) and (3), is between 5 and 10% of the total $\Delta H$, independently of the tidal range (not shown).


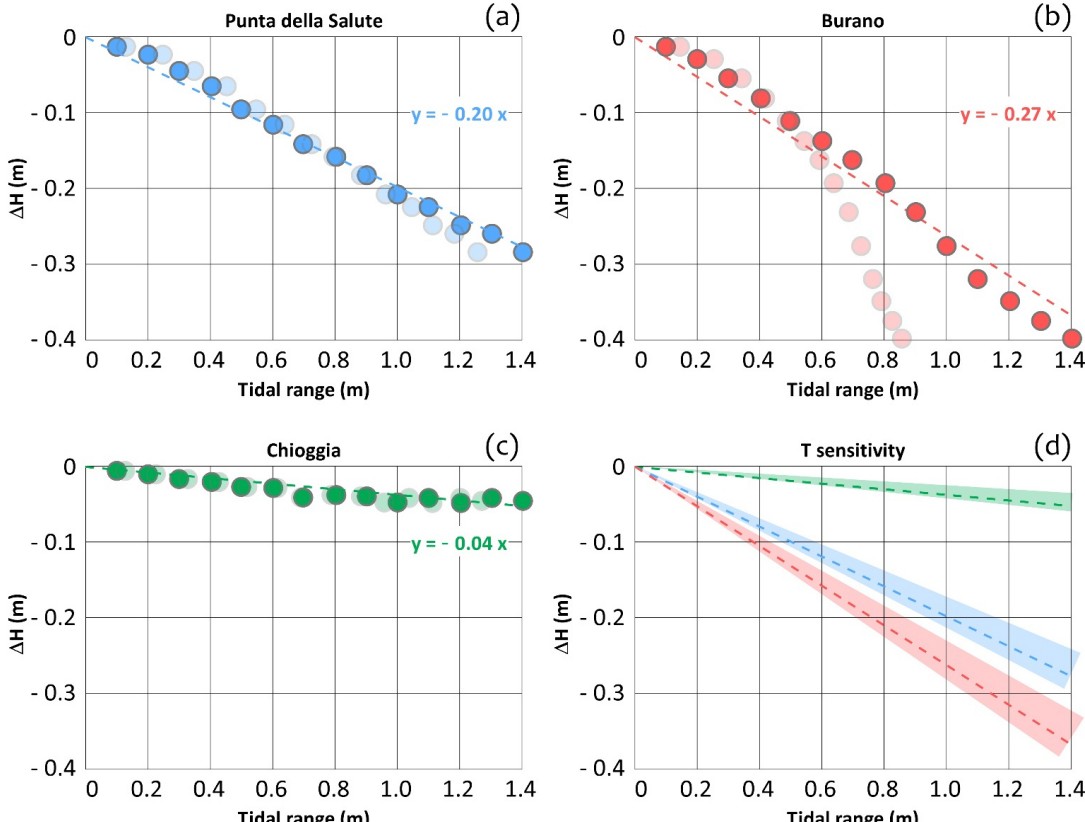

**Figure 3:** Synthetic sinusoidal tide; no wind condition (scenario (I)). Reduction of the sea level peak (ΔH) due to a partial use of the Mo.S.E. system (Lido inlet closed, PCL). (a-c) tidal semi-period of 6 hours. (a) Punta della Salute; (b) Burano; (c) Chioggia. Solid bullets refer to the seaward tidal range, opaque bullets to the tidal range computed at the specific gauge. (d) effect of the tide period on the slope of the linear regression (range 4 – 8 hours). Dashed lines refer to the relationships computed for a tidal semi-period of 6 hours at PS (blue, panel (a)), Burano (red, panel (b)) and Chioggia (green, panel (c)).

### 3.2 Scenario (II): the wind setup constraint

Mel et al. 2019a showed that the wind setup during the closure of the lagoon is significantly larger if compared to the unregulated condition. As the full use of the Mo.S.E. system could flood Chioggia during intense north-easterly winds (Mel et al., 2019a), it is crucial to investigate the specific effect of the PCL on wind setup. Figure 4 compares the relationship between tidal range and ΔH if Bora (45 °N) and Sirocco (165 °N) winds are superimposed to a tide of 6 hours period. Panels (a), (b)



and (c) illustrate the results at PS, Burano and Chioggia for a wind speed of 15 m/s. The relationship is still linear (black bullets and solid lines). If compared to scenario (I) (dashed lines), the slope of the regression does not change, but the intercept is not null. In case of wind, the PCL would produce an additional variation of the SL peak with respect to scenario (I) (namely, ΔHWIND, i.e., the distance between solid and dashed lines). ΔHWIND can be positive or negative and depends only on the characteristics of the wind (speed and direction) and not on the tidal range. Specifically, in case of Sirocco the PCL could be

detrimental, producing SL peaks even higher with respect to the unregulated condition (up to 0.1 m, Fig. 4a and 4b). Conversely, Bora, which is the prevailing wind in the Venice lagoon, would reduce the SL peak at all the three gauges if the PCL is implemented. This result is very important, as the PCL does not enhance the wind setup at Chioggia, as occurs when all the three inlets are closed. Notably, in case of Bora, the PCL enhances the SL gradient between PS at Chioggia, but only by reducing the SL at PS and not by increasing the SL at Chioggia. Panel (d) shows the relationship between wind speed and

ΔHWIND for Bora and Sirocco, confirming an additional reduction of the SLs in the whole lagoon in case of north-easterly winds and a SL increase in case of south-easterly winds. This phenomenon can be explained by the hydrodynamic effect of a wind blowing over Venice lagoon on the flow rate across the three inlets (see Zecchetto et al., 1997; Mel et al., 2019a). In an unregulated lagoon, when Bora wind is blowing, SL increases in the southern part of the lagoon and decreases in its northern part. Therefore, an additional water volume flows into the lagoon through the northern Lido inlet and out of the lagoon through

the southern Chioggia inlet. Conversely, Sirocco wind produces an additional volume entering the lagoon through the Chioggia inlet and an additional volume discharged toward the sea through the Lido inlet. In both the cases, the additional water volume flowing through the Lido inlet will disappear during the PCL. Therefore, in case of Sirocco, the net volume entering the lagoon will increase, enhancing the SLs in the lagoon. Conversely, Bora wind reduces the total water volume entering the lagoon, decreasing the SLs accordingly (Fig. 4d).

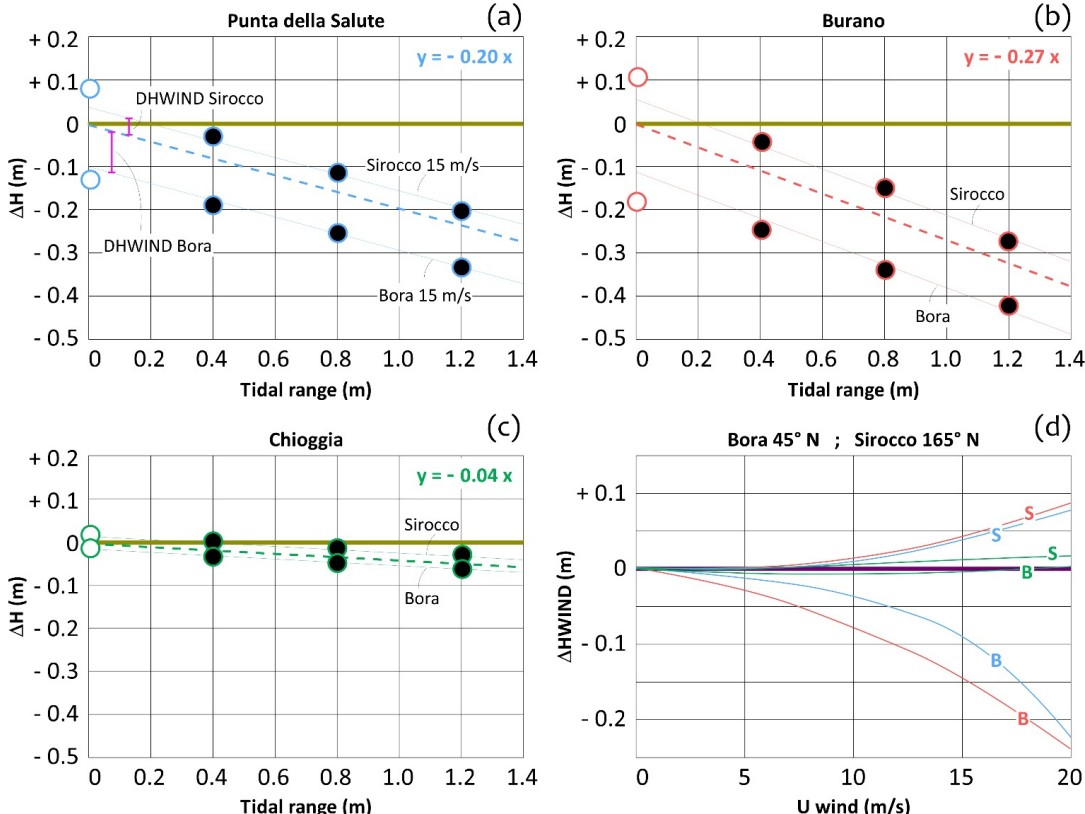

**Figure 4:** Synthetic sinusoidal tide, tidal semi-period of 6 hours, PCL. ΔH variation in case of wind (ΔH$_{WIND}$). (a) Punta della Salute; (b) Burano; (c) Chioggia. Dashed lines refer to the relationship obtained with no wind (scenario (I), Fig. 3); solid thin lines to scenario (II), with Sirocco and Bora wind of 15 m/s. White bullets refer to the simulations performed with no tide. (d) ΔH variation with respect to the no wind condition (ΔHWIND) computed at PS (blue lines), Burano (red) and Chioggia (green), ranging the speeds of Bora and Sirocco from 0 to 20 m/s.

### 3.3 Scenario (III): 2019 – 2020 dataset

The three linear relationships (1), (2) and (3) provided in Section 3.1 for estimating the ΔH referred to the seaward tidal range were verified considering data from 42 storm events occurred in the years 2019 and 2020 (Fig. 5). Results refer to the gauges of PS (blue), Burano (red) and Chioggia (green). Figure 5a and 5b compare the ΔH computed by the hydrodynamic model forced by the observed tidal and meteorological data (bullets) to the ΔH estimated by using the Eq. (1), (2) and (3) (dashed



lines, scenario (I)). The bullets nicely match the dashed lines confirming the robustness of the linear regression. The mean error considering all the events is 2 cm for PS, 4 cm for Burano and 1 cm for Chioggia. The reliability of the linear regression can be improved if some events are excluded from the comparison. Specifically, the most important discrepancies affect (a) events where a significant Bora (B) or Sirocco (S) wind blowed over the lagoon (U wind > 10 m/s) at the SL peak (black

bullets); (ii) events that involved two tidal cycles due to a prominent seiche wave in the Adriatic Sea, failing the hypothesis of sinusoidal tide (white bullets, see the example of 25 December 2019 in Fig. 5c); (iii) the exceptional storm occurred on 12 November 2019 (rhombus, see Cavaleri et al., 2020 for more details about the singularity of such event). Remarkably, Bora (B) and Sirocco (S) respectively enhance and reduce the $\Delta H$, as demonstrated in Section 3.2 (Fig. 4d). Figure 5c illustrates an example of the effect of the PCL on the SL at PS within the period 21-25 December 2019 (these events are labelled in Fig. 5a

as $21 - 25$). The five events falling within that period are characterized by different tidal and meteorological forcings. Specifically, on 21 December Sirocco reduces the $\Delta H$ by introducing a negative $\Delta HWIND$, i.e., the discrepancy between the purple line, which represents the SL computed by the model, and the grey horizontal line, the SL computed at the peak using Eq. (1). On 25 December, the tide signal was significantly altered by a longitudinal seiche affecting the Adriatic Sea, showing an inflection (highlighted by the orange circle in Fig. 5c) which reduces the benefit of the PCL. Notably, the events occurred

on 23 and 24 December, characterized by the highest tidal ranges (≥1.30 m), nicely fit the linear regression (Fig. 5a).
Results confirm the reliability of equations (1), (2) and (3), with particular reference to events not involving more than one tide cycle (which are very rare) and not affected by strong winds. Such equations can be conveniently applied to any surge based on the tidal and wind dataset (see Section 3.5).

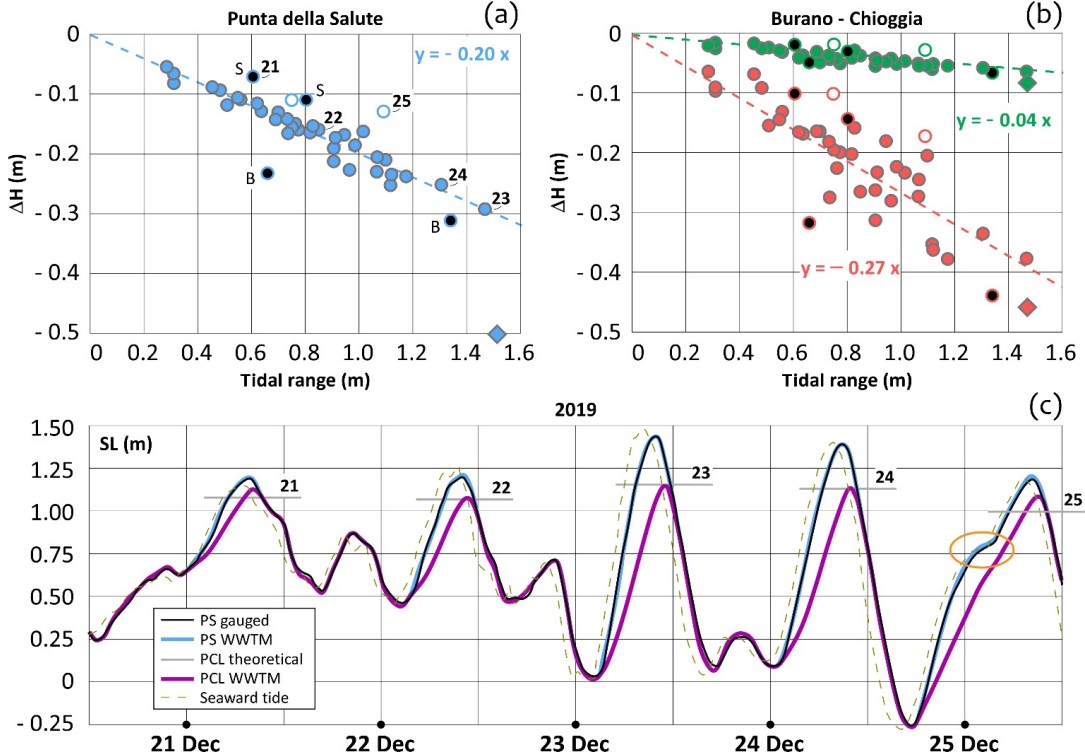

**Figure 5:** Dataset 2019 – 2020. SL peaks ≥ 1.10 m at PS (or at CNR platform during the tests performed at the Mo.S.E. barriers since October 2020). Relationship between tidal range and ΔH at (a) Punta della Salute (blue); (b) Burano (red) and Chioggia (green). Black bullets indicate the events characterized by strong Bora (B) or Sirocco (S); white bullets the events where two tidal cycles where merged due to a prominent seiche wave in the Adriatic Sea; rhombus the event of 12 November 2019. (c) timeseries 21 – 25 December 2019 at PS. SL observed (black line) and computed by means of the hydrodynamical model (PCL, purple; unregulated condition, blue). SLs computed at the peak by Eq. (1) are represented by the grey horizontal lines; mean seaward SL by the grey dashed line.

## 3.4 Flow rate trough the inlets

The impact of the PCL on the lagoon hydrodynamics is significant (Fig. 6). The PCL significantly affects the velocity field during the flood tide and vanishes the Lido sub-basin, causing an enlargement of the Malamocco sub-basin and a slightly different position of the Chioggia sub-basin (Fig. 6, black solid lines).

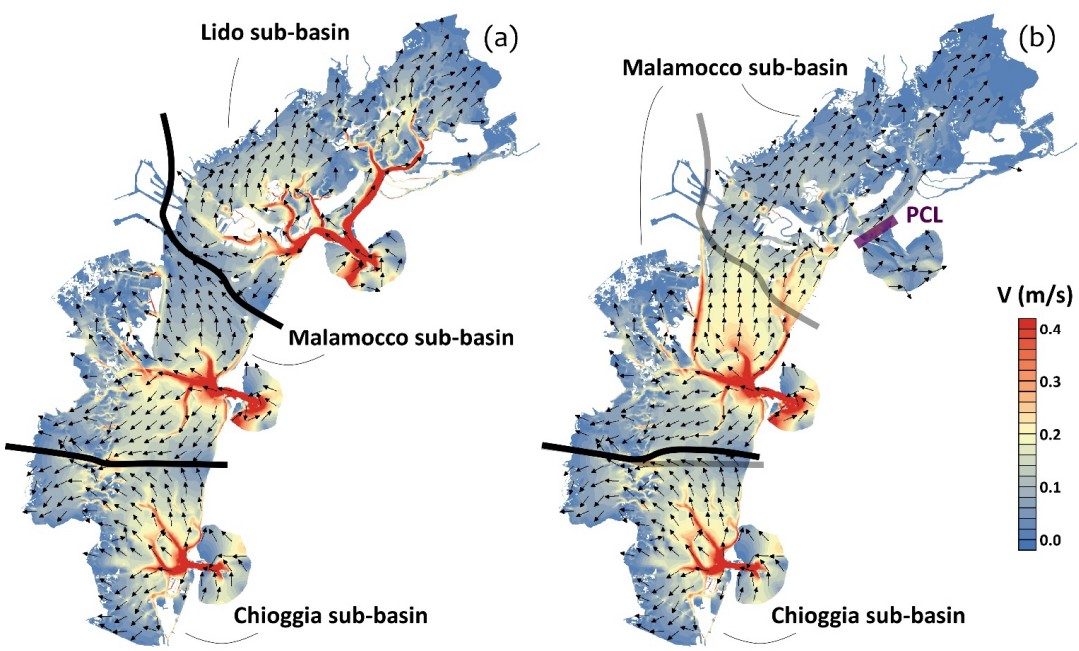


**Figure 6:** Sinusoidal tide, no wind (scenario I). Tidal range 0.8 m, tidal period 6 h. The two panels represent the 2-D velocity field and watersheds location (solid black lines) during at the 5th hours of the seaward flood tide. (a) unregulated lagoon (b) PCL.

The effects of the PCL on the Venice lagoon hydrodynamics have been addressed in terms of alteration of flow rates through the three inlets. For each inlet and tidal range belonging to the scenario (I), the incoming fluxes during the flood tide have been

compared to the conditions of an unregulated lagoon, in accordance with the method described in Section 2.3. Figure 7 shows the results for a tidal period of 6 hours. The comparison of the average and maximum flux evidences that the PCL enhances the flow rate through the Malamocco and Chioggia inlets for all the tidal ranges (Fig. 7a). The relationship is linear, with regression slopes of 1.14 (Malamocco inlet, yellow) and 1.02 (Chioggia inlet, green), for the maximum flow rate. Similar values (not shown) are obtained for the average flow rate. A wider difference is observed on the water volumes flowing into

the lagoon (Fig. 7b), as the PCL increases the duration of the flood tide due to lower internal SLs. Specifically, a volume increase of 22% is evidenced through the Malamocco inlet and 8% through the Chioggia inlet. Conversely, the total volume flowing into the lagoon through the three inlets is reduced by 21% (blue squares).


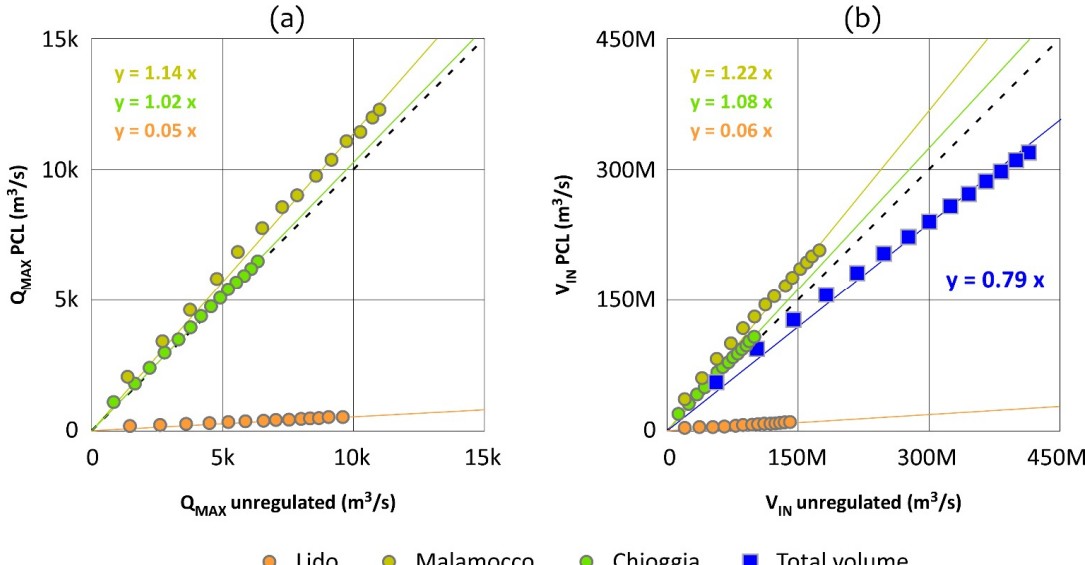

**Figure 7:** Sinusoidal tide, no wind (scenario I). Tidal range from 0.1 to 1.4 m, period of 6 hours. Comparison of maximum flow rate (a) and
water volume entering the lagoon (b) at the three inlets during the flood tide (x-axes unregulated lagoon, y-axes PCL).

These results indicate a modest effect of the PCL on the flux through the Malamocco and Chioggia inlets (i.e., the same
difference could be achieved by increasing of 0.1 m the tidal range in an unregulated lagoon).

### 3.5 Operating the Lido inlet under climate change

The linear relationship described in Section 3.1 allows to estimate the number of PCL capable to limit the SL below the
threshold at PS, Burano and Chioggia. The hindcast dataset of tidal and meteorological observations of the period 2000 – 2019
has been analysed. For investigating possible future scenarios, SLs were increased by adding different values of RSLR up to
+ 0.5 m (i.e., slightly exceeding the projected RSLR in northern Adriatic Sea for the year 2100 under the IPCC RCP 2.6
scenario, see Zanchettin et al., 2020) at steps of + 0.1 m, assuming that the mean SL does not significantly affect tidal range
and surge characteristics, as shown by several studies (Bondesan et al., 1995; Umgiesser and Matticchio, 2006; Mel et al.,
2013). Figure 8 shows the annual number of events over threshold. Specifically, orange bars represent the unregulated
condition, blue bars the residual events over threshold by implementing the PCL to all the events (unfiltered analysis).
Nevertheless, strong winds and events that involve multiple tidal cycles can reduce the benefit of the PCL, as shown in Section
3.2 and 3.3. Accordingly, in a second analysis some events have been excluded from the potential to implement the PCL

(filtered analysis, green bars). Specifically, the filtered analysis omitted: (i) the events of period greater than 8 hours, filtering possible flood tides that involve two or more tidal cycles (see Section 3.3); (ii) the events showing significant southern winds at the SL peak (i.e. wind speed $\geq$ 10 m/s and wind direction in the range 150° - 210° at the Laguna Nord gauge), filtering possible lower $\Delta$Hs due to the wind setup (i.e. negative $\Delta$HWIND, see Section 3.2); (iii) the events showing SL peaks $\geq$ 1.30 m at Diga Sud Chioggia, filtering possible water levels over threshold at Chioggia, as the small benefit of the PCL on Chioggia

SLs had been neglected. Green bars represent the residual annual number of events over threshold if the PCL is implemented only in the events which do not show the abovementioned characteristics (i) – (iii) (filtered analysis). The filtered analysis provides a residual number of events over threshold ranging from 33% to 38% of the total events for a RSLR up to + 0.40 m, and of 50% for a RSLR of + 0.50 m. These residual events should be faced by closing all the three inlets (full use of the Mo.S.E. system). Though the filters (i) – (iii) are very precautionary, a modest discrepancy between filtered and unfiltered

analysis is evidenced.

Although the uncertainties regarding climate change are still large, this study confirms the efficiency of the PCL as adaption strategy to extend the lifetime of the Mo.S.E. system. Results (Fig. 8) show that the PCL would have the same effect of increasing by 10 – 15 cm the safeguard threshold, keeping the frequency of the full use of the Mo.S.E. to an acceptable number for few decades, gaining precious time to verify the ongoing evolution of the climatic scenarios, narrowing the actual large

uncertainty ranges.

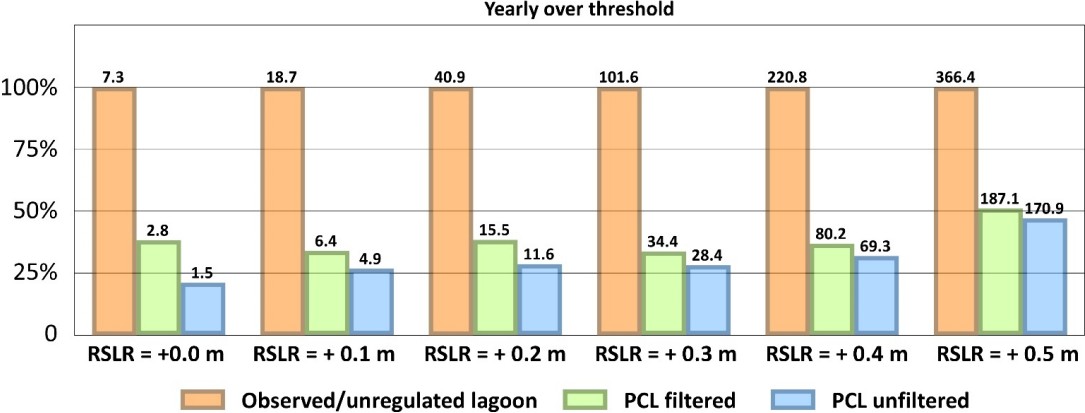

**Figure 8:** 2000-2019 dataset with a RSLR ranging from + 0.0 m (present condition) to +0.5 m. Comparison of the annual number of closures to the increasing RSLR. Orange bars illustrate the total number of SL peaks over threshold; green bars the residual events over threshold by implementing the PCL under the filtered scenario; blue bars the residual events if the PCL will be implemented during all the SL peaks.

This analysis focuses only on flood protection. As the PCL would significantly modify the lagoon hydrodynamics (Fig. 6), the long-term morphological and ecological consequences of such an operation strategy, which are beyond the scope of this work,




surely deserve further investigations. The optimal adaption and management strategy need to be carefully investigated and identified with the aim to balance flood protection with economic issues and the safeguarding of the lagoon ecosystem.

## 4 Conclusions

Structural measures, as well as high level of technical knowledge, are not necessarily the panacea for the long-term safety of the coast. Adaptation to climate change requires an effective and integrated management of coastal areas, with particular reference to the Venice lagoon, a World Heritage site enhancing threatened by flooding and erosion processes. Though the Mo.S.E. system has the potential to reduce the vulnerability of the Venice lagoon, serious questions arise whether it is beneficial to the morphology, the ecosystem, and to the wide range of uses of the lagoon. It is of paramount importance to

integrate the management of the barriers into a strategic context that addresses the social, economic and environmental issues raised in any coastal area. Accordingly, some measures should be undertaken to minimize the impacts of the operation of the Mo.S.E. system on water renewal, port activities and fishing industry. In order to reduce the number of simultaneous closures of the three inlets during flood events characterized by non-extreme SLs, the Malamocco and Chioggia inlets can be left open ensuring the flow of part of the water and the transit of vessels. Such solution requires to adapt the current management criteria

of the barriers. Toward this goal, the present study shows the results of a hydrodynamic investigation in which the partial use of the Mo.S.E. has been simulated by closing the only Lido inlet (PCL). Simulations were carried out considering both synthetic and realistic tide and wind scenarios. Results show that:

- a linear relationship was obtained between the tidal range of the seaward tide signal and the reduction of the sea level (SL) peak in the lagoon (ΔH). Specifically, the reduction has been estimated in 20% at Punta della Salute (Venice),
27% at Burano and 4% at Chioggia; the linear contribute (5-10%) of intragate infiltration has been accounted for. Tidal semi-period does not significantly affect the results;
- north-easterly winds would produce beneficial effects within all the lagoon, reducing the SL peaks. Conversely, south-easterly winds could produce higher SLs respect to an unregulated lagoon;
- events that involve multiple tidal cycles often show inflections of the tide signal, reducing the effectiveness of the
PLC;
- two third of the flood events can be effectively faced by the PCL under relative sea level rise scenarios up to + 0.4 m.

Besides this attempt to study some major hydrodynamic aspects related to the partial use of the Mo.S.E. barriers, many open issues remain to be analysed, as the possible long-term effects of such operation strategy on the bio-morphodynamic evolution of the Venice lagoon. A thorough cost-benefit analysis should be pursued to identify the optimal Mo.S.E. management strategy

supporting the resilience of Venice and its lagoon to high tides. The Venice lagoon test case points out the importance of



combining non-structural and structural measures to counteract flood risk in coastal areas, bearing in mind the need of investigating the long-term effects related to the strategies adopted to face present priorities.

**Acknowledgments.** Luca Carniello and Daniele Pietro Viero are gratefully acknowledged for fruitful discussions.

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

**Appendix A. The Mo.S.E. system testing phase**

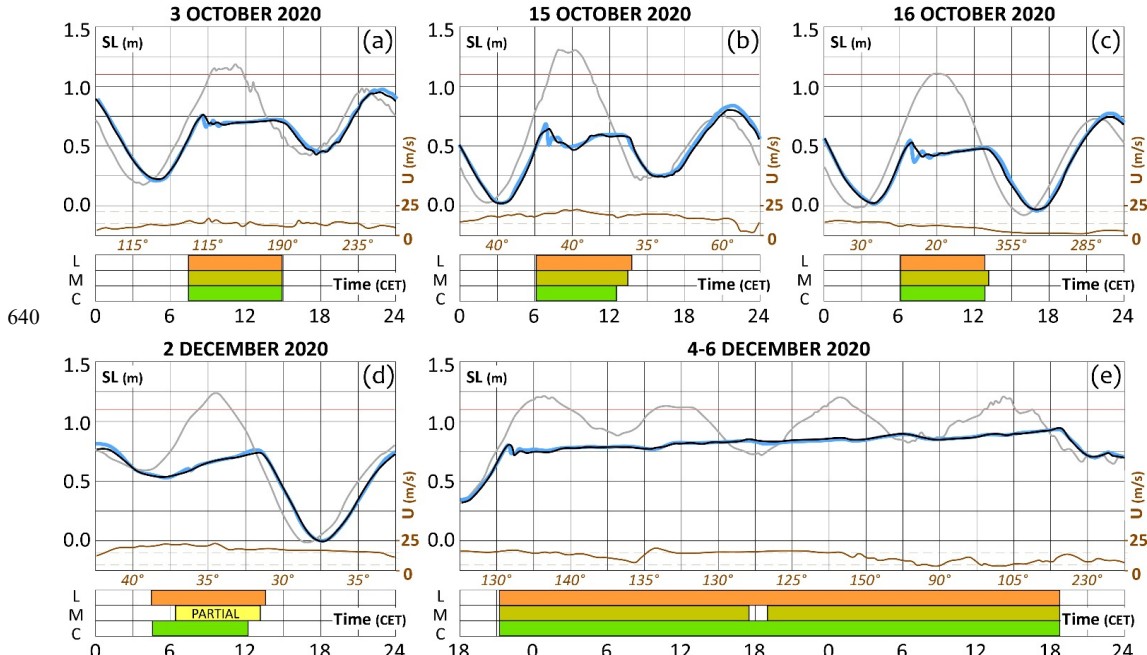




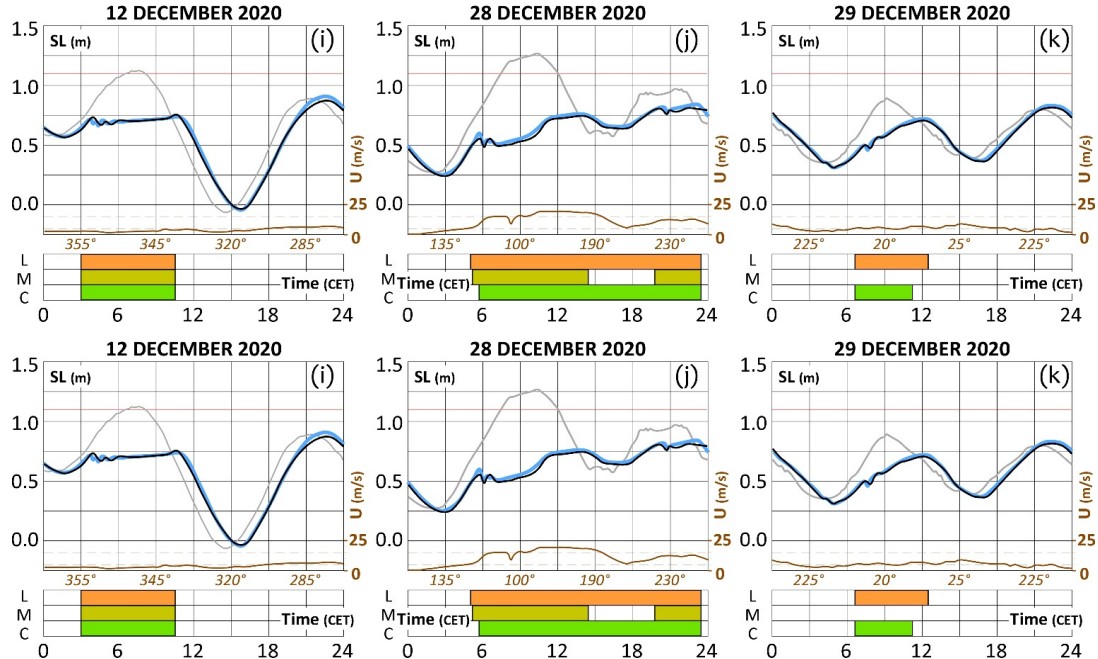


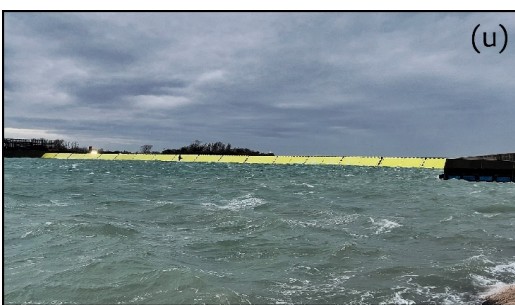
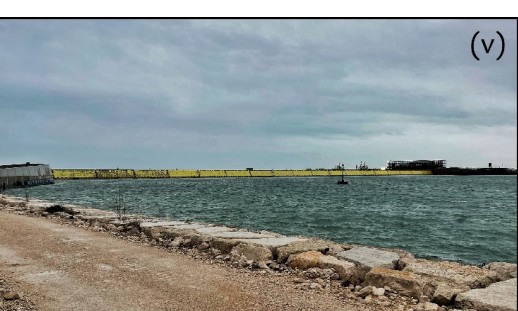

**Figure A.1: Events when the Mo.S.E. barriers have been raised under testing phase. (a) – (t) comparison between SLs observed at the CPSM gauge of PS (black lines) and computed by the WWTM model forcing the closure of the Mo.S.E. barriers (blue lines).**
**Grey and brown lines represent respectively the SLs and the wind speeds observed at the CPSM gauge of CNR platform. Red line represents the safeguard thresholds at PS (1.10 m). Bars indicate the periods when the barriers of Lido (orange), Malamocco (yellow) and Chioggia (green) have temporarily closed the respective inlets. (u) and (v) seaward and lagoonal view of the flap gates raised under the test of 28 December 2020 (photos by R.A. Mel).**

