# Peer review of "Exploring the partial use of the Mo.S.E. system as effective adaptation to rising flood frequency of Venice"

_Natural Hazards and Earth System Sciences, 2021_

## Author Comment (AC1)

Please note that in this rebuttal, *italics* refer to the text of the reviewer' comments, the detailed response is in **bold red**.

**AUTHOR RESPONSE TO REVIEWER #1:**

*Summary:*

*The manuscript describes a numerical study of the Mo.S.E. flood protection scheme in Venice. Specifically, the author assesses the effects of a Partial Closure of the lagoon exclusively involving the Lido inlet (PCL) and successfully showcases the potential of this novel operation concept. Although recent studies about this topic exist, the presented results further elucidate the influence of tidal range, wind setup and intra-gate infiltration, which complements the understanding of the city's capacity to adapt to increasing flood frequencies as a consequence of Climate Change.*

*General comments:*

*Although the hydrodynamic situation of the Venice lagoon is doubtlessly unique, the presentation of results and underlying methodology can readily be transferred to other cases of numerical modelling. The text is very well structured and the provided figures concisely summarize and illustrate all relevant findings. It was a pleasure reading this positive piece of work and, therefore, only minor revisions could be proposed that mainly address the style of presentation or mere Formalia.*

**We thank Reviewer #1 for the comments and suggestions that will help to improve the manuscript. Please, find below the author's answers.**

*Minor comments:*

*L72: "According to recent studies…" – it is arguable whether these studies are recent, but all of them seem to be anticipatory with regard to Mo.S.E.*

*L103: Figure 1 – a rectangular box around the Lido inlet may be added to define the extents of the blowup region more clearly*

*L104: "… physical, biogeochemical, and biological conditions…" – biological concerns seem to appear twice here*

*L109: "Nutrient and pollutants … from the drainage basin." – the location of this drainage basin and its discharge system may be of interest to the reader and could be included in Figure 1.*

*L144: Chapter 2.1.1 – this is the only third-order section in this chapter and its content could well be included in both the previous section 2.1 or in an individual section of second-order*

*L179: Appendix A – the figures generally attest a good resemblance between observed and simulated tidal curves, but could still benefit from additional quantitative justification, such as root mean square errors or correlation coefficients etc.*

*L228: "Results are not affected …" – they are affected, but not significantly*

*L228: Figure 3d – in a greyscale print, the regression lines can hardly be distinguished, which may be enhanced by different colour depths or name tags*

*L259-264: For the first time, the text loses some of its conciseness. Hydrodynamically, the lagoon simply becomes a one-ended basin.*

*L264: Figure 4d – again, lines in the fourth plot are hard to read in greyscale (and for people with colour vision deficiencies)*

*L282: "… the singularity of such event." – a short indication of the general nature of this event would be very helpful at this point*

*L322: "… same difference could be achieved by increasing of 0.1 m the tidal range …" – the origin of this quantity is not self-evident*

*L348: cf. comment on L322*

*L386: "… combining structural and non-structural measures …" – it is arguable whether the operation of the Mo.S.E. concept can be called an (individual) non-structural measure*

**The reviewer is right in all the minor comments she/he provided. All these suggestions will be carefully considered in the new version of the manuscript.**

*Formalia:*

*L65: „… independently flap gates …" – a word (reference of the adverb) seems to be missing*

*L68: "Works … begun in 2003." – participle tense should be checked*

*L70: "see Appendix A" – legend entries below figure (r) all contain "L" for Lido inlet*

*L161: "… a coupled wind wave-tide model …" – presumably a wind-wave tide model is meant*

*L195: "… all the events (n° 42) …" – presumably the total number of events was 48, which may better be expressed as "(N = 42)"*

*L202: "… the tidal dynamics has been reproduced …" grammatical number should be checked*

*L273: "… data from 42 storm events occurred in the years 2019 and 2020." – presumably "events that/which occurred"; further examples of this use of participles follow*

*L315: Figure 7b – both axes refer to volumes and accordingly would usually be measured in cubic meters ($m^3$)*

*L362: "… a World Heritage site enhancing threatened by flooding …" – presumably "increasingly threatened"*

*L378: "… higher SLs respect to an unregulated lagoon." – a word seems to be missing here*

*L380: "… reducing the effectiveness of the PLC" – presumably Partial Closure of the Lagoon involving the Lido inlet only (PCL).*

**The author is very grateful with the reviewer for noting all the Formalia that will help to improve the manuscript.**

---

## Author Comment (AC2)

Please note that in this rebuttal, *italics* refer to the text of the reviewer' comments, the detailed response is in **bold red**.

**AUTHOR RESPONSE TO REVIEWER #2:**

*GENERAL COMMENTS*

*This is a very pleasant and informative study. It sets a clear objective and exposes its results in a way that leaves little room for debate. A few typos and language imprecisions to be edited before final publication. I note a few (definitely not all) in the line-by-line comments, but overall the manuscript is a pleasant read. Some sentences could be divided in two for clarity.*

**The author thanks the reviewer for the positive evaluation of the manuscript and for the precious recommendations.**

*ABSTRACT*

*I feel like the abstract is slightly out of balance: strong focus on the rationale but vary succinct on the methods and results. In that sense it almost feels more like an introduction. Maybe shift the focus toward more methods/results.*

**The author agrees with the reviewer. In the new version of the manuscript the abstract will be corrected accordingly.**

*INTRODUCTION*

*Very well referenced and structured.*

**Thank you for the positive evaluation.**

*METHODS*

*Not much to say, this is clearly explained and based on a sound reasoning. The base model is also tried and tested.*

*In run (b): do you re-open the gates at any point during the ebb tide? If so it needs to be mentioned otherwise its seems like the tide is left to flow out of only the Malamocco and Chioggia inlet*

**Run (b) is used to determine the intragate infiltration. Run (c) is used to determine the optimal re-opening time of the Lido inlet (i.e., when the SL at Lido is lower in the sea). Accordingly, in both runs (b) and (c) the tide begins to flow out through only the Malamocco and Chioggia inlets, while Lido inlet is kept closed.**

*In run (d): you do mean that each gate is opened separately on the condition that the water level difference is 0?*

**As during the flood tide under the PCL the (lagoonal) SL difference between the Treporti and San Nicolò inlet is not negligible, each gate is opened separately when the SL difference between the sea and the lagoon sides is null.**

*My only question would be: how does the timing of gate closure affect your results? Probably not within the scope of this paper, but worth mentioning for further contributions.*

**A slight advance/delay of the closure of the Lido inlet (up to one hour) would not significantly affect the effect of the PCL. Conversely, it is more important (but operationally easier, i.e., a null SL difference to gauge and a faster manoeuvre of the gates during the re-opening phase) to identify the correct re-opening time.**

*RESULTS*

*Very clearly presented. The results fit the objectives determined in the introduction.*

**Thank you for the positive evaluation.**

*FIGURE COMMENTS*

*Fig1: a rough indication of the delimitations of the sub-basins would add interesting information without adding too much clutter to the figure. Not a necessary modification though as these delimitations are seen in figure 6. I missed the legend for the control sections 1-3, since their description comes quite a bit later in the text.*

**In the new version of the manuscript the figure will be corrected accordingly. Thank you for noting.**

*Fig3: (d) The effect of the tidal semi-period is not immediately visible. While it is visible that it is represented by the filled polygons, some text to indicate the effect of the shorter period vs. the longer one could help the reader.*

**The small effect of the tidal (semi-) period within the range of study is not proportional to the tidal period itself. The maximum effect of the PCL on the SL peak reduction will be achieved with a tidal period of about 5.5 hours. In the new version of the manuscript this result will be reported. Thank you for noting.**

*Fig4: This is a very nice and clear figure, it is immediately comprehensible.*

**Thank you.**

*Fig5: idem*

**Thank you.**

*LINE-BY-LINE SUGGESTIONS*

*l9. Detail a bit more for readers (impact of sea level rise and subsidence)*

*l12. "sediment flushing"*

*l15: with respect => compared to*

*l21. does the population do the adapting or is it the flood management system?*

*l33. delete either "prevented" or "fixed"*

*l48. You do not mention changes in storm regimes. I'm aware the change in storm regime is debated but a mention of publications on the subject could address a reader's questions on the subject.*

*l71. "rose" => "raised".*

*l174. I wonder if is worth mentioning that the bathymetry inside the lagoon (not at the inlets) predates 2012.*

*l230. "contribute"=>"contribution"*

*l278. "(a)=>"(i)"*

*l290. Even though events involving more than 1 tidal cycle are rare, a discussion on the effect of these rare events would be interesting in this case: what would be the decision criteria to close the gate for long periods?*

**In the new version of the manuscript all these line-by-line suggestions will be carefully taken in account. Thank you.**

As concern the comment on line 290, the optimal operation on the gates for the events that involve more than one tidal cycle should be carefully identified. As a first note, the PCL will not be effective, as stated in the manuscript. The decision criteria to close the (three) inlets for long periods is fully debated in the following paper, cited in the present manuscript:

Mel, R., Carniello, L., and D'Alpaos, L.: How long the Mo.S.E. barriers will be effective in protecting all urban settlements within the Venice lagoon? The wind setup constraint. Coastal engineering, 168, 103923, doi:10.1016/j.coastaleng .2021.103923, 2021a.